# TRIM2 Selectively Regulates Inflammation-Driven Pathological Angiogenesis without Affecting Physiological Hypoxia-Mediated Angiogenesis

**DOI:** 10.3390/ijms25063343

**Published:** 2024-03-15

**Authors:** Nathan K. P. Wong, Emma L. Solly, Richard Le, Victoria A. Nankivell, Jocelyne Mulangala, Peter J. Psaltis, Stephen J. Nicholls, Martin K. C. Ng, Christina A. Bursill, Joanne T. M. Tan

**Affiliations:** 1Vascular Research Centre, Lifelong Health Theme, South Australian Health and Medical Research Institute, Adelaide, SA 5000, Australia; nwon9940@alumni.sydney.edu.au (N.K.P.W.); emma.solly@sahmri.com (E.L.S.); richard.le@flinders.edu.au (R.L.); victoria.nankivell@sahmri.com (V.A.N.); jocelyne.mulangala@heartfoundation.org.au (J.M.); peter.psaltis@sahmri.com (P.J.P.); christina.bursill@sahmri.com (C.A.B.); 2Faculty of Medicine and Health, The University of Sydney School of Medicine, Camperdown, NSW 2050, Australia; mkcng@med.usyd.edu.au; 3Department of Cardiology, St. Vincent’s Hospital, Darlinghurst, NSW 2010, Australia; 4Adelaide Medical School, Faculty of Health and Medical Sciences, University of Adelaide, Adelaide, SA 5005, Australia; 5College of Medicine and Public Health, Flinders University, Adelaide, SA 5042, Australia; 6Heart Foundation, Brisbane, QLD 4000, Australia; 7Victorian Heart Institute, Monash University, Clayton, VIC 3800, Australia; stephen.nicholls@monash.edu; 8Department of Cardiology, Royal Prince Alfred Hospital, Camperdown, NSW 2050, Australia

**Keywords:** inflammation, ischemia, neovascularization, HIF-1α, eNOS

## Abstract

Angiogenesis is a critical physiological response to ischemia but becomes pathological when dysregulated and driven excessively by inflammation. We recently identified a novel angiogenic role for tripartite-motif-containing protein 2 (TRIM2) whereby lentiviral shRNA-mediated TRIM2 knockdown impaired endothelial angiogenic functions in vitro. This study sought to determine whether these effects could be translated in vivo and to determine the molecular mechanisms involved. CRISPR/Cas9-generated *Trim2*^−/−^ mice that underwent a periarterial collar model of inflammation-induced angiogenesis exhibited significantly less adventitial macrophage infiltration relative to wildtype (WT) littermates, concomitant with decreased mRNA expression of macrophage marker *Cd68* and reduced adventitial proliferating neovessels. Mechanistically, TRIM2 knockdown in endothelial cells in vitro attenuated inflammation-driven induction of critical angiogenic mediators, including nuclear HIF-1α, and curbed the phosphorylation of downstream effector eNOS. Conversely, in a hindlimb ischemia model of hypoxia-mediated angiogenesis, there were no differences in blood flow reperfusion to the ischemic hindlimbs of *Trim2*^−/−^ and WT mice despite a decrease in proliferating neovessels and arterioles. TRIM2 knockdown in vitro attenuated hypoxia-driven induction of nuclear HIF-1α but had no further downstream effects on other angiogenic proteins. Our study has implications for understanding the role of TRIM2 in the regulation of angiogenesis in both pathophysiological contexts.

## 1. Introduction

Angiogenesis is the process in which new blood vessels are formed from pre-existing vessels. It is crucial in many physiological contexts, such as in wound healing, and as an adaptive response to hypoxia and ischemia [1]. However, imbalance in angiogenic regulation can cause deleterious effects, as it accelerates inflammation-driven pathologies, as seen in atherosclerosis and cancer [2,3]. New vessels provide additional conduits for the delivery of inflammatory cells and cytokines that promote atherosclerotic plaque development and rupture. They also deliver the oxygen and nutrients necessary to sustain tumor growth and serve as potential routes for metastatic spread [4].

Angiogenesis-associated conditions are highly prevalent globally, with cardiovascular disease (CVD) and cancer among the leading causes of morbidity and mortality worldwide. Current anti-angiogenic agents are limited, as they can interfere with physiological angiogenic processes, while pro-angiogenic therapies can potentially exacerbate chronic inflammation and inadvertently precipitate tumorigenesis [5,6]. Given the critical role of angiogenesis across such diverse pathologies, any agent capable of differentially modulating angiogenesis in a context-specific manner would be of great therapeutic value.

We previously identified a novel angiogenic role for tripartite motif-containing protein 2 (TRIM2) [7]. Lentiviral short hairpin (sh)RNA knockdown of TRIM2 impaired endothelial cell tubule formation in both hypoxia and inflammatory conditions in vitro [7]. We have also shown that TRIM2 knockdown attenuates the ability of human coronary artery endothelial cells (HCAECs) to migrate and proliferate in response to hypoxic and inflammatory stimuli. However, whether these effects are translated in vivo, and what the molecular mechanisms are underlying these, remains unknown.

In this study, we used CRISPR/Cas9-generated homozygous *Trim2* null (*Trim2*^−/−^) mice to evaluate the functional importance of TRIM2 in two well-validated models of pathological inflammation-driven angiogenesis and physiological hypoxia-mediated angiogenesis, namely the periarterial cuff and hindlimb ischemia models, respectively. In *Trim2*^−/−^ mice, we report markedly attenuated infiltration of adventitial macrophages in response to femoral artery cuff placement, when compared to wildtype (WT) littermates concomitant with a reduction in mRNA levels of the macrophage marker cluster of differentiation 68 (*Cd68*). Mechanistically, we show that TRIM2 knockdown in human coronary artery endothelial cells (HCAECs) attenuates the induction of key mediators involved in the classical inflammation-driven angiogenic signaling pathway, including nuclear translocation of hypoxia-inducible factor (HIF)-1α and phosphorylation of downstream mediator endothelial nitric oxide synthase (eNOS).

In contrast, we find no significant differences in blood flow reperfusion despite a reduction in proliferating neovessels and arterioles in the ischemic hindlimbs of *Trim2*^−/−^ and WT mice. In vitro, while the hypoxia-mediated induction of HIF-1α was tempered by TRIM2 knockdown, further downstream activation of angiogenic signaling proteins were unaffected. These findings collectively highlight a novel role for TRIM2 in the regulation of inflammation-driven angiogenesis and delineate the mechanistic basis for these effects. We propose TRIM2 to be a potential therapeutic target for diseases driven by pathological angiogenesis, unlimited by the usual adverse effects associated with inhibiting physiological angiogenesis.

## 2. Results

### 2.1. Plasma Glucose and Lipid Concentrations Are Not Affected by TRIM2 Knockdown

Deletion of *Trim2* from the tissues of *Trim2*^−/−^ mice was confirmed by qPCR, with only 7.0 ± 4.6% and 1.7 ± 0.5% of residual *Trim2* expression detected in the gastrocnemius muscle and liver tissues of *Trim2*^−/−^ mice, respectively (Appendix A Appendix A). All WT and *Trim2*^−/−^ animals were monitored regularly pre- and post-operatively until the conclusion of each study. This included the measurement of daily weights, which provided an opportunity to observe the mice for any major phenotypic differences that might develop. We observed no obvious differences noted with respect to general neurological or motor function, nor were there any clear differences in cardiovascular health. There were no differences in total body weights at the conclusion of the periarterial cuff or hindlimb ischemia studies or the weights of various individual organs between WT and *Trim2*^−/−^ mice (Appendix A Appendix A). As part of the phenotypic evaluation of *Trim2*^−/−^ mice, plasma glucose and lipid levels were determined. These metabolic parameters were invariably lower in *Trim2*^−/−^ mice relative to WT mice, though no statistically significant differences were observed (Table 1).

### 2.2. Trim2 Deletion Inhibits Infiltration of Adventitial Macrophages and Attenuates Angiogenic Responses to Inflammation In Vivo

In the periarterial cuff study, CD68^+^ macrophage infiltration into the adventitia of cuffed arteries was markedly reduced in *Trim2*^−/−^ mice (33.9 ± 10.2% vs. WT: 100.0 ± 27.4%, *p* < 0.05, Figure 1a). Furthermore, *Trim2*^−/−^ mice had fewer CD31^+^ vessels in the adventitia of cuffed arteries (73.0 ± 12.8% vs. WT: 100.0 ± 10.7%, Figure 1b), though this difference did not reach statistical significance (*p* = 0.1221). Notably, *Trim2*^−/−^ mice had significantly fewer Ki67^+^CD31^+^ proliferating neovessels (40.5 ± 10.5% vs. WT: 100.0 ± 16.4%, *p* < 0.01, Figure 1c). However, no differences were observed in the presence of CD34^+^ endothelial tip cells (Figure 1d). The intima-to-media ratio, a measure of inflammation-driven neointima formation, was similar between WT and *Trim2*^−/−^ mice (Figure 1e).

Together with reduced macrophage infiltration, there was attenuated induction of key inflammatory markers at 24 h post-surgery in the cuffed arteries of *Trim2*^−/−^ mice. Firstly, comparison between WT and *Trim2*^−/−^ mRNA levels of cuffed arteries showed a significant decrease in *Cd68* mRNA expression in the *Trim2*^−/−^ animals (55.9 ± 13.0% vs. WT: 100.0 ± 12.6%, *p* < 0.05, Figure 2a), consistent with the CD68^+^ macrophage staining. However, no differences were observed in *Ccl2* (Figure 2b) and *Rela* (Figure 2c) levels. We also assessed if global Trim2 knockout attenuates the extent of cuff-induced inflammatory response. When compared to their respective non-cuffed control arteries, there was a 42-fold increase in *Cd68* mRNA levels in the cuffed arteries of WT mice (*p* < 0.0001, Figure 2d). Cuff placement also induced Cd68 expression in *Trim2*^−/−^ mice; however, the extent of stimulation was less pronounced (30-fold, *p* < 0.001). We observed a similar pattern with *Ccl2* mRNA levels such that cuff placement induced a 200-fold increase in *Ccl2* in WT mice while there was a 150-fold induction in *Trim2*^−/−^ mice (Figure 2e). WT and *Trim2*^−/−^ had similar levels of *Rela* induction (Figure 2f).

### 2.3. Inflammation-Induced Activation of Angiogenic Signaling Mediators Is Attenuated by TRIM2 Knockdown In Vitro

To understand the mechanistic basis for the effects observed in vivo, we examined the modulation of several key angiogenic signaling mediators in vitro following lentiviral shRNA knockdown of TRIM2 in HCAECs. Comparison of TNFα-stimulated cells alone showed that nuclear NF-κB p65 protein levels was 30% lower in shTRIM2 cells (70.1 ± 29.0%, Figure 3a) when compared to shControl cells (100.0 ± 26.1%), but this did not reach statistical significance (*p* = 0.4952). However, nuclear HIF-1α protein levels were significantly lower in shTRIM2 cells (32.9 ± 8.3% vs. shControl: 100.0 ± 25.4%, *p* < 0.05, Figure 3b). No differences were observed in either PHD3 protein levels (Figure 3c). Interestingly, while no differences were observed in VEGFA protein levels (Figure 3d), TRIM2 knockdown in vitro augmented VEGFR2 activation (142.4 ± 13.4% vs. shControl: 100.0 ± 10.5%, *p* < 0.05, Figure 3e). No differences were observed in p38 MAPK phosphorylation (Figure 3f); however, eNOS phosphorylation was significantly attenuated (shTRIM2: 63.5 ± 9.3% vs. shControl: 100.0 ± 8.5%, *p* < 0.05, Figure 3g).

We also compared the extent of TNFα-induced inflammatory response when compared to respective baseline (No TNFα) controls. In shControl-transduced cells, inflammatory stimulation with TNFα significantly increased nuclear NF-κB p65 (*p* < 0.05, Figure 3h). While nuclear NF-κB p65 levels were also higher in TNFα-stimulated shTRIM2 cells, this did not reach statistical significance (*p* = 0.1999) when compared to its respective non-stimulated control. Strikingly, nuclear HIF-1α levels were significantly reduced in shTRIM2 cells when compared to TNFα-stimulated shControl cells (*p* < 0.05, Figure 3i). TNFα also significantly induced protein levels of PHD3 (*p* < 0.01, Figure 3j) and VEGFA (*p* < 0.05, Figure 3k) in shControl cells. However, these inflammatory-driven inductions were not seen with TRIM2 knockdown. VEGFR2 phosphorylation was reduced in response to TNFα stimulation in both shControl and shTRIM2 cells (Figure 3l). No differences were observed with p38 MAPK phosphorylation irrespective of conditions (Figure 3m). TRIM2 knockdown significantly reduced eNOS activation compared to both unstimulated shTRIM2 cells and TNFα-stimulated shControl cells (*p* < 0.05 for both, Figure 3n).

### 2.4. Trim2 Deletion Does Not Affect Angiogenic Responses to Ischemia In Vivo

In the hindlimb ischemia study, there were no differences between *Trim2*^−/−^ and WT mice in their capacity for blood flow reperfusion, as monitored by longitudinal laser Doppler imaging (Figure 4a). WT and *Trim2*^−/−^ mice were indistinguishable in their motor functions and the appearance of their distal limbs throughout the study.

Angiogenic responses to ischemia were further assessed histologically in the distal gastrocnemius muscle of the ischemic hindlimbs (Figure 4b). In *Trim2*^−/−^ mice, the density of CD31^+^ neovessels relative to the number of myocytes was increased (133.5 ± 9.5%, *p* < 0.05) when compared to that of WT mice (Figure 4c). However, the density of α-SMA^+^ arterioles relative to the number of myocytes was not different between the ischemic hindlimbs of *Trim2*^−/−^ and WT mice (Figure 4d). Interestingly, the number of Ki67^+^CD31^+^ proliferating neovessels was significantly decreased in *Trim2*^−/−^ mice (51.6 ± 7.9% vs. WT: 100.0 ± 15.7%, *p* < 0.05, Figure 4e). We also observed a decrease in Ki67^+^α-SMA^+^ arterioles in *Trim2*^−/−^ mice (56.4 ± 9.3% vs. WT: 100.0 ± 15.4%, *p* < 0.05, Figure 4f).

### 2.5. TRIM2 Knockdown Attenuates Hypoxia-Mediated Induction of Nuclear HIF-1α, PHD3 and VEGFA but Not Downstream Signaling Pathways In Vitro

In vitro, comparison of shTRIM2 and shControl cells exposed to hypoxia showed that TRIM2 knockdown significantly reduced nuclear HIF-1α protein levels (31.8 ± 9.3% vs. shControl: 100.0 ± 25.3%, *p* < 0.05, Figure 5a), with no differences observed in PHD3, VEGFA and activation of VEGFR2, p38 MAPK and eNOS (Figure 5b–f). We then compared the extent of hypoxic induction when compared to respective baseline controls. Under hypoxic conditions, nuclear HIF-1α levels were significantly reduced in shTRIM2 cells (44 ± 13%, *p* < 0.05, Figure 5g) when compared to hypoxia-stimulated shControl cells (138 ± 35%). Hypoxia also significantly increased PHD3 (*p* < 0.05, Figure 5h) and VEGFA (*p* < 0.05, Figure 5i) protein levels in shControl cells relative to their respective normoxia controls. However, these hypoxia-driven inductions were not observed in shTRIM2 cells. Meanwhile, VEGFR2, p38 MAPK and eNOS phosphorylation was unaffected by TRIM2 knockdown in HCAECs in hypoxia (Figure 5j–l).

## 3. Discussion

Dysregulated angiogenesis crucially underpins a wide range of chronic and debilitating diseases including atherosclerotic CVD and cancer. TRIM2 has emerged as a promising novel target that may differentially modulate both inflammation-driven pathological angiogenesis and hypoxia-stimulated physiological angiogenesis, as our previous studies have demonstrated impaired endothelial tubule formation in HCAECs with TRIM2 knockdown in vitro [7]. Here, we report markedly reduced adventitial macrophage infiltration following *Trim2* deletion in a murine periarterial cuff model of inflammation-driven angiogenesis, concomitant with reduced proliferating adventitial neovessels and attenuated the induction of the inflammatory response. Correspondingly, we find that TRIM2 knockdown in HCAECs suppresses the TNFα-driven induction of several classical angiogenic mediators, particularly nuclear HIF-1α and reduced activation of the eNOS angiogenic signaling pathway. *Trim2* deletion, however, did not alter the capacity for blood flow reperfusion nor the extent of neovascularization in the murine hindlimb ischemia model despite a reduction in proliferating neovessels and arterioles. While TRIM2 knockdown in vitro suppressed the hypoxia-driven stimulation of nuclear HIF-1α, it did not affect downstream expression and activation of pro-angiogenic signaling pathways.

Our findings, particularly from the periarterial cuff model and the mechanistic studies, are consistent with our previous work showing the inhibition of inflammation-induced endothelial tubule formation after TRIM2 knockdown [7]. The reduction in CD68^+^ macrophage infiltration into the inflamed arteries of *Trim2*^−/−^ mice suggests that TRIM2 may be involved in broader mechanisms of inflammatory activation, which enhance vessel growth by stimulating a wealth of pro-angiogenic growth factors and mediators [8]. The concomitant reduction in *Cd68* mRNA levels in the cuffed arteries of *Trim2*^−/−^ mice further support the idea that TRIM2 plays a key role in regulating inflammation-driven pathological angiogenesis, particularly in the early stages of macrophage recruitment to the site of injury. The reduction in proliferating adventitial neovessels in the cuffed arteries of *Trim2*^−/−^ mice is reflective of this. This was associated with a trend towards reduced total neovessels. Had the study duration been extended beyond 3 weeks, it could be hypothesized that a larger reduction in adventitial neovessels would be observed, as the blunted macrophage response would lead to fewer pro-angiogenic factors being released. No differences were observed in the presence of CD34^+^ endothelial tip cells. The intima-to-media ratio, though, which assesses the formation of a thickened neointima as an ‘outside-in’ response to adventitial inflammation, was not affected by *Trim2* deletion, indicating that the reduction in adventitial neovessels was a specific effect on angiogenesis and not a consequence of the development of a smaller neointimal or media. Furthermore, recent clinical studies have identified TRIM2 as a potential oncogene in human cancer cell lines including colorectal carcinoma, epithelial ovarian carcinoma and osteosarcoma [9,10,11]. These observations fit with a postulated role for TRIM2 in inflammation-driven angiogenesis, a hallmark of cancer development and progression.

Our findings also suggest a potential mechanistic pathway by which TRIM2 may be directing angiogenic responses to inflammation. Direct comparisons of TNFα-stimulated cells in vitro showed a significant reduction in nuclear HIF-1α levels and eNOS activation. Intriguingly, when compared to their respective baseline unstimulated controls, we found that the extent of inflammatory induction of nuclear NF-κB, PHD3 and VEGFA were less pronounced in shTRIM2 cells. Under stimulation with cytokines like TNFα, it is possible that TRIM2, functioning as a ubiquitin ligase [12], may contribute to the stabilization and nuclear translocation of the transcription factors NF-κB and HIF-1α, likely by promoting proteasomal degradation of their cytosolic inhibitors, such as PHD3 in the latter case [13]. These transcription factors, in turn, promote VEGFA expression, which activates endothelial cell migration and tubule formation through numerous intracellular pathways [14,15], including the phosphorylation of p38 MAPK and eNOS. NF-κB p65 and HIF-1α may also stimulate angiogenesis downstream of TRIM2 by upregulating inflammatory cytokines and chemokines like CCL2, leading to the recruitment and activation of macrophages that help to potentiate the inflammation-driven angiogenic response.

It is not clear whether TRIM2 mediates VEGFA-related effects via VEGFR2, as only a modest decrease in Tyr^1175^ phosphorylation was seen following TRIM2 knockdown. TRIM2 could be targeting alternative VEGFR2 phosphorylation sites. While there was no change in VEGFR2 phosphorylation at Tyr^1175^, it is possible that other key tyrosine sites like Tyr^801^, Tyr^1054^ or Tyr^1059^ may be involved, each of which may activate a distinct set of signal transduction mechanisms and cellular responses [14]. Specifically, VEGFR2 phosphorylation at Tyr^801^ also contributes to Akt-dependent eNOS activation and nitric oxide release from endothelial cells [16,17]. Another potential target of TRIM2 is neuropilin-1 (NRP1), one of the key co-receptors for VEGFR2. NRP1 is highly expressed in endothelial cells and neurons and can bind to both VEGFA and the class 3 semaphorins, a family of axonal guidance proteins, thus forming a key link between angiogenesis and neurogenesis [15,18]. Given the association of TRIM2 with axonal outgrowth and development [12,19], and now angiogenesis, it is possible that NRP1 and/or its semaphorin ligands may be involved in regulating angiogenic function by TRIM2. Further studies examining a possible link between NRP1 and TRIM2 would be useful to clarify the mechanistic pathway.

Consistent with our previous work which demonstrated impairment of hypoxia-stimulated tubule formation in vitro with TRIM2 knockdown [7], *Trim2* deletion reduced the number of proliferating neovessels and arterioles in the ischemic tissue in vivo. Interestingly, there was an increase in total CD31^+^ vessels in the ischemic hindlimbs of *Trim2*^−/−^ mice, suggestive of a potential negative feedback loop. In the late stages of angiogenesis, vessel pruning occurs, whereby capillaries disintegrate, to facilitate mature vessel formation during tissue remodeling [20]. We therefore postulate that the changes seen with increased capillary density yet reduced number of proliferating neovessels may be indicative of vessel pruning at this late stage post-ischemia. Furthermore, revascularization in the hindlimb ischemia model is primarily facilitated through arteriogenesis [21]. No differences were observed in overall arteriolar density, which is consistent with the lack of change seen in blood flow reperfusion. However, while there were significant changes at a cellular/tissue level, this did not seem to impact the recovery of blood flow reperfusion to the ischemic hindlimb. These incongruous findings may reflect inherent angiogenic compensatory mechanisms in vivo that may be activated in response to *Trim2* deletion, perhaps starting early in embryonic development and thereby rendering *Trim2* redundant. Mechanistically, while the hypoxia-driven increase in nuclear HIF-1α was attenuated in TRIM2-deficient HCAECs, downstream angiogenic signaling was not altered. The paucity of effects on these intracellular mediators may explain the lack of differences between WT and *Trim2*^−/−^ mice in their angiogenic responses to hypoxia. This could prove clinically useful, as anti-TRIM2 therapies may be developed to suppress pathological inflammatory angiogenesis, without the adverse effects of impairing hypoxia-driven physiological angiogenesis.

The discordant in vitro and in vivo findings may also reflect the activation of compensatory angiogenic mechanisms in response to *Trim2* deletion in vivo. Future studies could explore inducible and endothelial cell-specific *Trim2* silencing to exclude such effects as angiogenic responses may be countered by *Trim2* deletion in other cell types, like vascular smooth muscle cells, pericytes and immune cells [22]. It is also plausible that TRIM2 may target non-classical pathways downstream of VEGFA to confer its angiogenic effects in hypoxia, or it could be modulating other angiogenic factors like the fibroblast growth factors or angiopoietins. Future studies could evaluate a broader range of signaling targets to better elucidate the mechanistic basis by which TRIM2 may be modulating endothelial responses to hypoxia.

Overall, we have shown, for the first time, that TRIM2 is functionally important in regulating pathological angiogenic responses to inflammation in vivo, via modulation of classical angiogenic mediators HIF-1α, NF-κB p65, and VEGFA and downstream targets of VEGFA. Given that TRIM2 has no effect on physiological ischemia-driven angiogenesis, targeted TRIM2 inhibition could prove therapeutically useful for diseases driven predominantly by pathological angiogenesis including atherosclerosis and cancer, without the adverse effects of inhibiting physiological angiogenesis.

## 4. Materials and Methods

### 4.1. Animal Studies

All experimental procedures were conducted with approval from the SAHMRI Animal Ethics Committee (#SAM335) and conformed to the Australian Code for the Care and Use of Animals for Scientific Purposes (National Health and Medical Research Council, Australia). A *Trim2*^−/−^ mouse line was generated by the South Australian Genome Editing facility using a CRISPR/Cas9 approach. In brief, Cas9 protein was injected into C57BL/6J murine embryos along with two guide RNA sequences. These guide RNAs were designed such that non-homologous end joining of the DNA following CRISPR/Cas9 activity would result in excision of a DNA fragment containing exon 2 of *Trim2*, leading to a frameshift in the coding sequence and an early stop codon in exon 3. The founder male carrying this mutant *Trim2* allele was back-crossed to wildtype (WT) female C57BL/6J mice, generating identical heterozygous offspring which were subsequently crossed to generate homozygous *Trim2* knockout (*Trim2*^−/−^) mice. Male wildtype (WT) and *Trim2*^−/−^ mouse littermates were housed in a temperature and humidity-controlled environment under a 12 h light/dark cycle with ad libitum access to water and standard mouse chow. They underwent surgery at 8 weeks of age.

### 4.2. Plasma Glucose and Lipid Analyses

Plasma glucose concentrations were determined using a glucometer (Accu-Chek^®^ Performa, Roche, Basel, Switzerland), while total plasma and HDL cholesterol concentrations were measured enzymatically (439-17501, Wako Diagnostics, Richmond, VA, USA). HDL cholesterol concentrations were determined following polyethylene glycol precipitation of apoB-containing lipoproteins, while LDL cholesterol concentrations were calculated by subtracting HDL from total cholesterol concentrations. Triglyceride concentrations were determined using a colorimetric assay (290-63701, Wako Diagnostics).

### 4.3. Periarterial Cuff Model

The femoral periarterial cuff model is an established model of inflammation-driven neointima formation and adventitial angiogenesis [23,24], processes which are known to contribute to atherosclerotic plaque development. A non-occlusive 2 mm length of polyethylene cuff was placed around the left femoral artery to trigger a localized inflammatory response, while a sham operation was performed on the right femoral artery as a parallel control. The animals were sacrificed 21 days post-surgery by overdose of isoflurane and intracardiac puncture, followed by perfusion with phosphate-buffered saline (PBS) via the left ventricle. The femoral arteries (complete with cuff) were excised for histochemical analyses.

Excised femoral arteries were fixed in 10% (*v*/*v*) formalin for 24 h then embedded in 3% (*w*/*v*) agarose prior to tissue processing and paraffin embedding. Angiogenic responses to cuff placement were assessed via immunohistochemistry on 5 μm sections, probing for CD68 (Bio-Rad, Hercules, CA, USA, Cat# MCA1957GA, RRID:AB_324217) to assess macrophage infiltration and CD31 (Abcam, Cambridge, UK, Cat# ab28365, RRID:AB_726365) to detect adventitial vessels. Proliferating neovessels were determined by co-staining tissue sections with proliferation marker Ki-67 (Thermo Fisher Scientific, Waltham, MA, USA, Cat# 14-5698-82, RRID:AB_10854564) and CD31. Endothelial tip cells were determined by staining sections with CD34 (Abcam, Cat# ab8158, RRID:AB_306316). Masson’s trichrome staining was performed with a Trichrome Stain Kit (ab150686, Abcam) to assess intima-to-media ratio as a measure of neointimal responses to inflammatory stimulation. All histological sections were photographed with a Zeiss Axio Scan.Z1 Digital Slide Scanner (Carl Zeiss Microscopy, Oberkochen, Baden-Württemberg, Germany), and image analysis was performed using Image-Pro Premier software (v9.0.4, Media Cybernetics, Rockville, MD, USA).

An additional cohort of mice underwent the same procedure and were sacrificed 24 h post-surgery for gene expression analysis. Total RNA was isolated from the femoral arteries with TRI^®^ reagent (Sigma-Aldrich, St. Louis, MO, USA) and quantitated spectrophotometrically. Then, 200 ng of total RNA was reverse transcribed using the iScript cDNA synthesis kit (Bio-Rad). Quantitative real-time PCR was performed for *Cd68* (F: 5′-GGACAGCTTACCTTTGGATTCAA-3′; R: 5′-CTGTGGGAAGGACACATTGTATTC-3′), *Ccl2* (F: 5′-GCTGGAGCATCCACGTGTT-3′; R: 5′-ATCTTGCTGGTGAATGAGTAGCA-3′), NF-κB p65 (*Rela*, forward [F]: 5′-AGTATCCATAGCTTCCAGAACC-3′; reverse [R]: 5′-ACTGC-ATTCAAGTCATAGTCC-3′) and *36B4* (F: 5′-CAACGGCAGCA-TTTATAACCC-3′; R: 5′-CCCATTGATGATGGAGTGTGG-3′). Relative gene expression was calculated using the ^ΔΔ^Ct method, normalized to *36B4* and WT non-cuffed arteries.

### 4.4. Hindlimb Ischemia Model

The hindlimb ischemia model is a well-validated model of physiological angiogenesis in response to tissue ischemia [25]. Hindlimb ischemia was induced by ligation and excision of the left superficial and deep femoral arteries, along with the left femoral vein down to the saphenous artery. A sham procedure was performed on the contralateral hindlimb as a parallel control. Hindlimb blood reperfusion was determined by laser Doppler imaging (moorLDI2-IR, Moor Instruments, Devon, UK), performed prior to and immediately following surgery, then at days 1, 3, 6, 8 and 10 post-surgery. Animals were sacrificed 10 days post-surgery by isoflurane overdose and intracardiac puncture, and the gastrocnemius muscles of both hindlimbs were collected for histological analyses.

Gastrocnemius muscles from both ischemic and non-ischemic hindlimbs were OCT-embedded and frozen on dry ice. Sections were taken across the medial plane of the gastrocnemius muscle (anterior distal hindlimb). This region is known to provide the most consistent and uniform responses to ischemic induction [25,26,27]. To histologically assess angiogenic responses to ischemia, immunofluorescence was performed on 5 µm tissue sections, staining with CD31 (Abcam, Cat# ab28364, RRID:AB_726362) to detect neovessels, α-smooth muscle actin (α-SMA, Sigma-Aldrich, Cat# F3777, RRID:AB_476977) to detect arterioles and laminin (Millipore, Burlington, MA, USA, Cat# MAB1905, RRID:AB_94392) to detect myocytes. Proliferating neovessels and arterioles were determined by co-staining tissue sections with Ki-67 (Thermo Fisher Scientific, Cat# 11-5698-82, RRID:AB_11151330) and either CD31 or α-SMA, respectively. Images were taken using an Eclipse Ni-E fluorescent microscope (Nikon Instruments, Tokyo, Japan). CD31^+^ neovessels and α-SMA^+^ arterioles were quantified using CellProfiler software (www.cellprofiler.org, accessed on 9 January 2023, Broad Institute of MIT and Harvard, Boston, MA, USA), while the myocytes were manually quantified using ImageJ (https://imagej.net/ij/, accessed on 9 January 2023, National Institutes of Health, Bethesda, MD, USA).

### 4.5. Lentiviral shRNA Knockdown of TRIM2 In Vitro

Human coronary artery endothelial cells (HCAECs, Cell Applications, San Diego, CA, USA) were cultured in MesoEndo Cell Growth Medium (212-500, Cell Applications) and used at passages 3–4. HCAECs were seeded at 5 × 10^4^ cells/well in 6-well plates and cultured at 37 °C and 5% CO_2_ overnight. The cells were exposed to 1 × 10^4^ infectious units (IFU)/mL of lentiviral particles containing shRNA against TRIM2 (shTRIM2) or a random control sequence (shControl) for 24 h in the presence of polybrene. Transduced HCAECs were trypsinized, counted and seeded at a density of 1.5 × 10^5^ cells/well and 8 × 10^4^ cells/well for the inflammation and hypoxia experiments, respectively. HCAECs were then either incubated for 4.5 h with 0.6 ng/mL TNFα (to mimic inflammation) or for 6 h at 5% CO_2_ and 1.2% O_2_ balanced with N_2_ (to mimic hypoxia). To measure phosphorylated proteins, HCAECs were stimulated with 10 ng/mL recombinant human VEGF_165_ protein (R&D Systems) 15 min prior to harvest. Nuclear proteins were isolated from cell lysates using the NE-PER^®^ Nuclear and Cytoplasmic Extraction kit (Thermo Fisher Scientific). Whole-cell protein lysates were extracted using RIPA buffer [7,28]. Each experiment was performed at least four times independently with triplicates for each condition.

### 4.6. Protein Expression

Nuclear and whole-cell protein extracts were subjected to Western blot analysis and probed with primary antibodies for NF-κB p65 (Abcam, Cat# ab16502, RRID:AB_443394), HIF-1α (Novus Biologicals, Centennial, CO, USA, Cat# NB100-105, RRID:AB_10001154), PHD3 (Novus, Cat# NB100-303, RRID:AB_10003302), VEGFA (Abcam, Cat# ab46154, RRID:AB_2212642), phosphorylated (Tyr^1175^) VEGFR2 (Cell Signaling Technology, Danvers, MA, USA, Cat# 2478, RRID:AB_331377), total VEGFR2 (Cell Signaling Technology, Cat# 2479, RRID:AB_2212507), phosphorylated (Thr^180^/Tyr^182^) p38 MAPK (Cell Signaling Technology, Cat# 4511, RRID:AB_2139682), total p38 MAPK (Cell Signaling Technology, Cat# 8690, RRID:AB_10999090), phosphorylated (Ser^1177^) eNOS (BD Biosciences, Franklin Lakes, NJ, USA, Cat# 612393, RRID:AB_399751) and total eNOS (BD Biosciences, Cat# 610297, RRID:AB_397691). Even protein loading was confirmed with lamin B1 (Abcam, Cat# ab16048, RRID:AB_443298) for nuclear fractions or α-tubulin (Abcam, Cat# ab40742, RRID:AB_880625) for whole-cell lysates.

### 4.7. Statistics

Data are expressed as mean ± SEM. Comparisons were made using unpaired Student’s *t*-tests or two-way ANOVA followed by post hoc analysis using Bonferroni’s multiple comparison tests. Significance was set at a two-sided *p* < 0.05.

## 5. Conclusions

In conclusion, we have shown, for the first time, that TRIM2 is functionally important in regulating pathological angiogenic responses to inflammation. We found that *Trim2*^−/−^ mice that underwent a periarterial collar model of inflammation-induced angiogenesis exhibited significantly less adventitial macrophage infiltration, concomitant with decreased *Cd68* mRNA levels. *Trim2*^−/−^ mice also had reduced adventitial proliferating neovessels. Mechanistically, our in vitro findings show that TRIM2 knockdown inhibits nuclear HIF-1α translocation and eNOS phosphorylation (Figure 6). Given that TRIM2 appears to have limited bearing on physiological ischemia-driven angiogenesis, TRIM2-directed therapies may represent safer alternatives to current anti-angiogenic strategies for the treatment of atherosclerotic CVD, cancer and chronic rheumatological conditions.

## Figures and Tables

**Figure 1 ijms-25-03343-f001:**
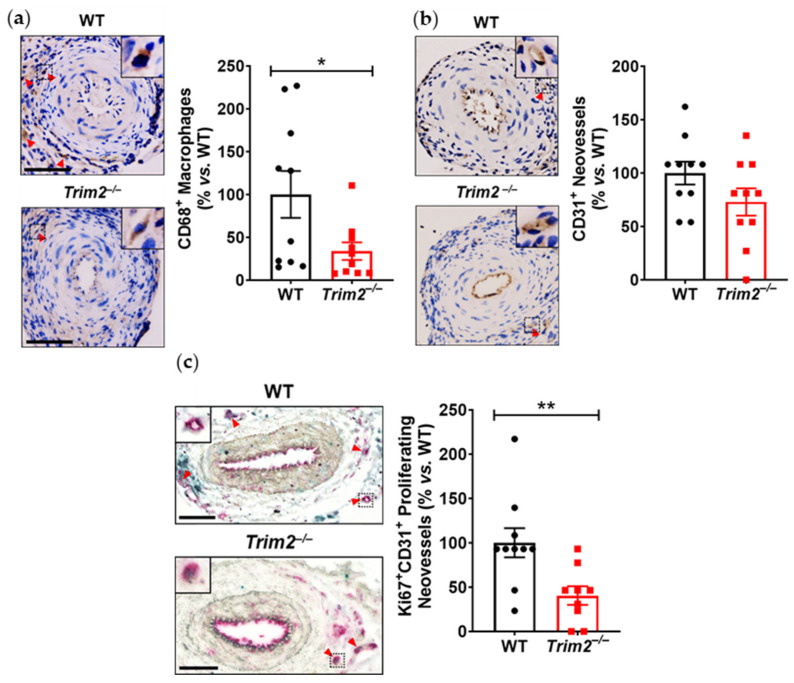
Trim2 deletion inhibits inflammatory-driven adventitial macrophage infiltration in vivo. A non-occlusive polyethylene cuff (2 mm) was placed around the left femoral arteries of wildtype (WT) and *Trim2*^−/−^ mice (*N* = 10/group) for 21 days to trigger localized inflammatory responses. Femoral arteries were sectioned for immunohistochemical detection of adventitial (**a**) CD68^+^ macrophages (brown staining, red arrowheads), (**b**) CD31^+^ neovessels (brown staining, red arrowheads), (**c**) Ki67^+^CD31^+^ proliferating neovessels (brown and red dual staining, red arrowheads) and (**d**) CD34^+^ endothelial tip cells (brown staining, red arrowheads). (**e**) Masson’s trichrome staining was performed to assess intima-to-media ratio. Representative images of cuffed artery sections from WT and *Trim2*^−/−^ mice were taken at 40× magnification. Scale bars: 50 μm. Results are mean ± SEM. * *p* < 0.05, ** *p* < 0.01 using unpaired Student’s *t*-test.

**Figure 2 ijms-25-03343-f002:**
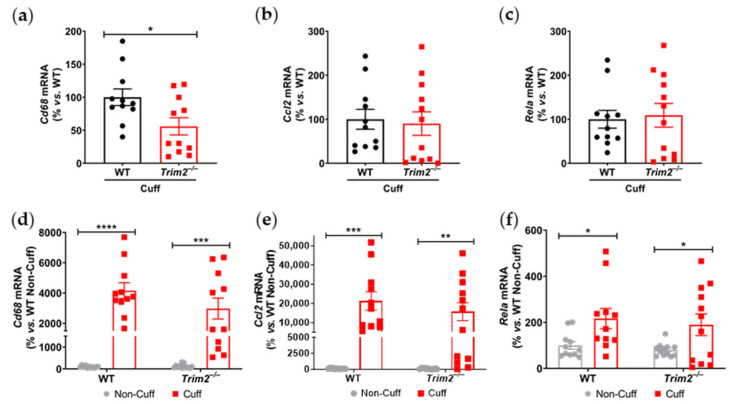
*Trim2* deletion attenuates angiogenic responses to inflammation in vivo. A non-occlusive polyethylene cuff (2 mm) was placed around the left femoral arteries of wildtype (WT) and *Trim2*^−/−^ mice (*N* = 10/group) for 24 h to trigger localized inflammatory responses. (**a**) *Cd68*, (**b**) *Ccl2* and (**c**) NF-κB p65 (*Rela*) mRNA levels in cuffed arteries of WT and *Trim2*^−/−^ mice, normalized using the ^ΔΔ^Ct method to *36B4* and WT cuffed arteries. (**d**) *Cd68*, (**e**) *Ccl2* and (**f**) NF-κB p65 (*Rela*) mRNA levels in non-cuffed and cuffed arteries of WT and *Trim2*^−/−^ mice, normalized using the ^ΔΔ^Ct method to *36B4* and WT non-cuffed arteries. Results are mean ± SEM. * *p* < 0.05, ** *p* < 0.01, *** *p* < 0.001, **** *p* < 0.0001 using unpaired Student’s *t*-test or two-way ANOVA with Bonferroni’s post hoc analysis.

**Figure 3 ijms-25-03343-f003:**
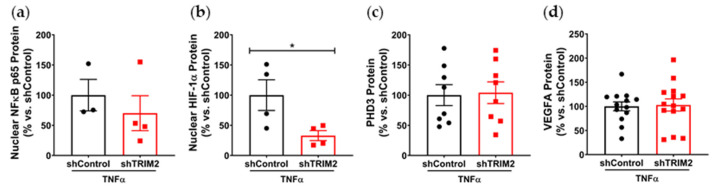
Inflammation-induced activation of angiogenic signaling mediators is attenuated by TRIM2 knockdown in vitro. Lentiviral-transduced shControl and shTRIM2 HCAECs were incubated without and with TNFα (0.6 ng/mL, 4.5 h). Protein levels of (**a**) nuclear NF-κB p65, (**b**) nuclear HIF-1α, (**c**) PHD3, (**d**) VEGFA, (**e**) Phospho:Total VEGFR2, (**f**) Phospho:Total p38 MAPK and (**g**) Phospho:Total eNOS, presented as percent change relative to TNFα-treated shControl cells. Protein levels of (**h**) nuclear NF-κB p65, (**i**) nuclear HIF-1α, (**j**) PHD3, (**k**) VEGFA, (**l**) Phospho:Total VEGFR2, (**m**) Phospho:Total p38 MAPK and (**n**) Phospho:Total eNOS, presented as percent change relative to No TNFα shControl cells. Dotted lines separate noncontiguous lanes from the same gel. The cropped blots are used in the figure, and the full blots are presented in Appendix A Appendix A. Each experiment was conducted at least three times independently with triplicates for each condition. Results are mean ± SEM. * *p* < 0.05, ** *p* < 0.01 *** *p* < 0.001 using unpaired Student’s *t*-test or two-way ANOVA with Bonferroni’s post hoc analysis.

**Figure 4 ijms-25-03343-f004:**
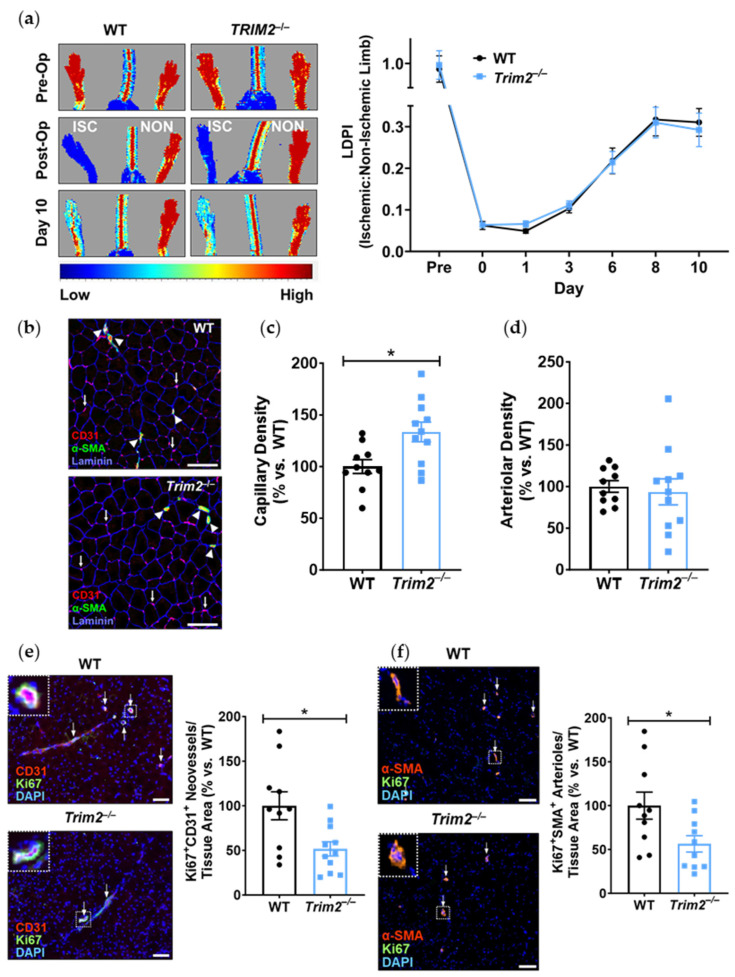
Trim2 does not affect angiogenic responses to ischemia in vivo. Wildtype (WT) and *Trim2*^−/−^ mice (*N* = 10/group) underwent ligation and excision of the left femoral artery and vein. (**a**) Blood flow reperfusion was measured by laser Doppler imaging over 10 days. Representative images show high (red) to low (blue) blood flow in the ischemic (ISC) and non-ischemic (NON) hindlimbs. Laser Doppler perfusion index (LDPI) was calculated as the ratio of flow in the ISC:NON hindlimbs. (**b**) Representative sections taken across the medial plane of the ischemic gastrocnemius muscles of WT and *Trim2*^−/−^ mice taken at 20× magnification showing CD31^+^ neovessels (red/purple staining, arrows), α-SMA^+^ arterioles (green staining, arrowheads) and laminin-stained basement membrane of the muscle fibers (blue staining). (**c**) The density of CD31^+^ neovessels per myocyte, normalized to WT mice. (**d**) The density of α-SMA^+^ arterioles per myocyte, normalized to WT mice. Immunofluorescent co-staining was performed to detect (**e**) Ki67^+^CD31^+^ proliferating neovessels and (**f**) Ki67^+^α-SMA^+^ proliferating arterioles. Scale bars: 100 μm. Results are mean ± SEM. * *p* < 0.05 using unpaired Student’s *t*-test.

**Figure 5 ijms-25-03343-f005:**
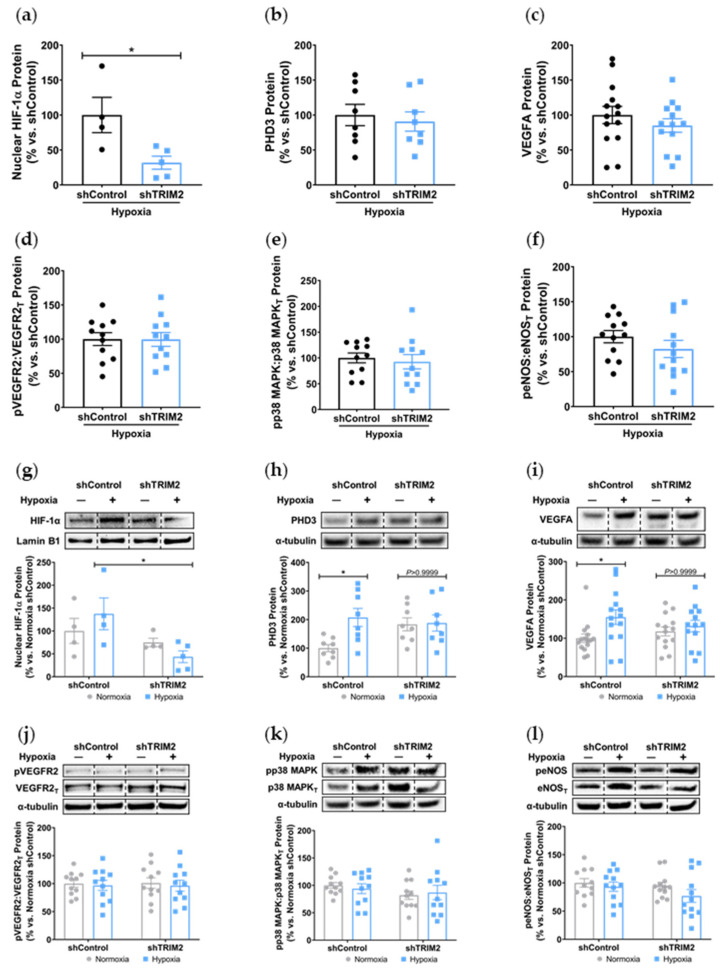
TRIM2 knockdown in vitro attenuates hypoxia-driven induction of nuclear HIF-1α, PHD3 and VEGFA but not downstream signaling pathways. Lentiviral-transduced shControl and shTRIM2 HCAECs were incubated in normoxia or hypoxia (1.2% O_2_, 6 h). Protein levels of (**a**) nuclear HIF-1α, (**b**) PHD3, (**c**) VEGFA, (**d**) Phospho:Total VEGFR2, (**e**) Phospho:Total p38 MAPK and (**f**) Phospho:Total eNOS, presented as percent change relative to hypoxia-stimulated shControl cells. Protein levels of (**g**) nuclear HIF-1α, (**h**) PHD3, (**i**) VEGFA, (**j**) Phospho:Total VEGFR2, (**k**) Phospho:Total p38 MAPK and (**l**) Phospho:Total eNOS, presented as percent change relative to normoxia shControl cells. Dotted lines separate noncontiguous lanes from the same gel. The cropped blots are used in the figure, and the full blots are presented in Appendix A Appendix A. Each experiment was conducted at least three times independently with triplicates for each condition. Results are mean ± SEM. * *p* < 0.05 using unpaired Student’s *t*-test or two-way ANOVA with Bonferroni’s post hoc analysis.

**Figure 6 ijms-25-03343-f006:**
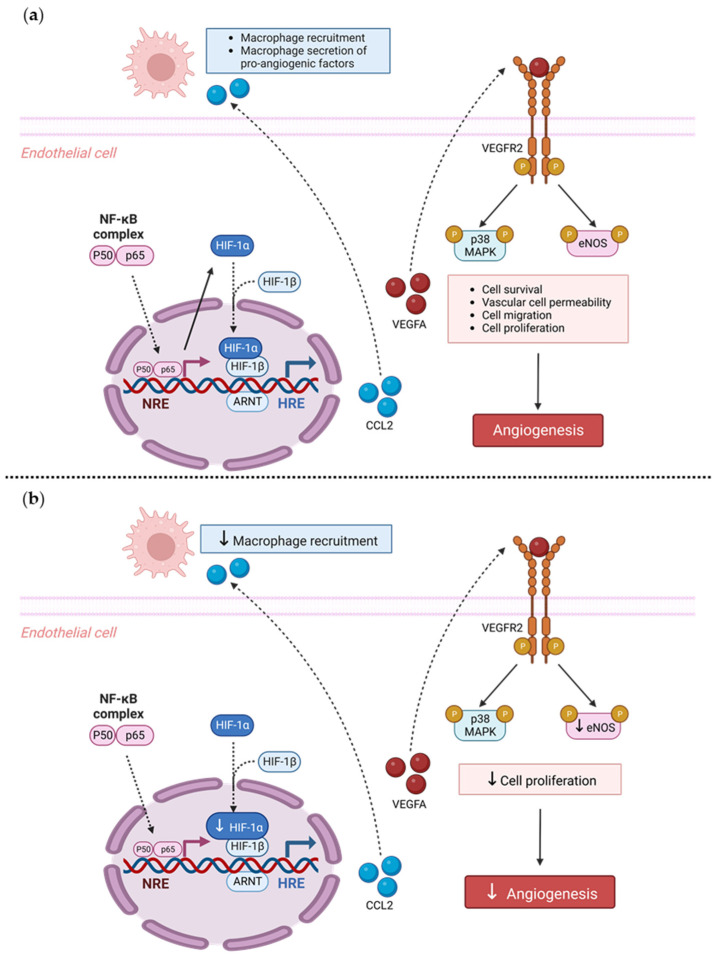
Proposed mechanistic role of TRIM2 in inflammatory-driven angiogenesis. (**a**) In response to an inflammatory stimulus such as TNFα, nuclear translocation of the key inflammatory transcription factor NF-κB occurs. The NF-κB complex comprises the p50/p65 subunits, forming a dimer that translocates into the nucleus. The p50/p65 dimer binds to NF-κB response elements (NREs) and upregulates a range of inflammatory and angiogenic targets including HIF-1α and VEGFA. Nuclear HIF-1α (nHIF-1α) dimerizes with HIF-1β, and together they bind to hypoxia-response elements (HREs), leading to upregulation of VEGFA, among many other pro-angiogenic genes. VEGFA binds to the VEGFR2 receptor, resulting in autophosphorylation of tyrosine residues in the cytoplasmic domain of VEGFR2. This leads to the phosphorylation (denoted by P circles) and activation of downstream signaling mediators including eNOS and p38 MAPK, resulting in angiogenesis. Additionally, NF-κB activates pro-inflammatory cytokines such as CCL2, which facilitate the recruitment of macrophages. Macrophages contribute to inflammatory-driven angiogenesis by secreting pro-inflammatory angiogenic factors. The solid arrows indicate activation, and dashed arrows indicate translocation. (**b**) Our study found that *Trim2*^−/−^ mice that underwent a periarterial collar model of inflammation-induced angiogenesis exhibited significantly less adventitial macrophage infiltration, concomitant with decreased *Cd68* mRNA levels. *Trim2*^−/−^ mice also had reduced adventitial proliferating neovessels. Mechanistically, our in vitro findings show that TRIM2 knockdown inhibits nuclear HIF-1α translocation and eNOS phosphorylation. Figure adapted from [28]. Created with BioRender.com (accessed on 9 January 2023).

**Table 1 ijms-25-03343-t001:** Plasma glucose and lipids measured in WT and *Trim2*^−/−^ mice.

Plasma Parameter	WT (*N* = 23)	*Trim2*^−/−^ (*N* = 23)
Glucose (mM)	14.5 ± 0.3	14.0 ± 0.4
Total cholesterol (mg/dL)	256.3 ± 7.4	239.7 ± 7.2
HDL cholesterol (mg/dL)	122.2 ± 4.1	118.2 ± 4.4
LDL cholesterol (mg/dL)	134.1 ± 6.0	121.5 ± 5.7
Triglycerides (mg/dL)	100.4 ± 7.6	97.0 ± 6.2

Data are shown as mean ± SEM.

## Data Availability

The original contributions presented in the study are included in the article/Appendix A. Further inquiries can be directed to the corresponding author.

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
