# Peer review of "TRIM2 Selectively Regulates Inflammation-Driven Pathological Angiogenesis without Affecting Physiological Hypoxia-Mediated Angiogenesis"

_ijms, 2024, doi:10.3390/ijms25063343_

Round 1

Reviewer 1 Report

Comments and Suggestions for Authors

In the present study, Wong et al. tested the role of TRIM2 as an inhibitor of angiogenesis in mice using two established models that distinctly represent inflammatory or ischemia-driven angiogenesis. The authors showed that knockdown of TRIMP2 reduced the formation of new blood vessels and macrophage infiltration in inflamed tissue, while the angiogenic response after hypoxia was almost unaffected. In experiments mimicking inflammation and hypoxia in vitro, silencing TRIM2 in human coronary artery endothelial cells primarily attenuated the induction of inflammatory mediators such as HIF-1α and partially eNOS. The authors concluded that TRIM2 has a key role in the regulation of angiogenesis.

The paper is well written and the authors provide detailed experiments on the critical role of TRIMP2 in pathological angiogenesis. I have only a few minor comments.

Minor comments

o   How were the samples fixed before paraffin embedding?

o   Explain HCAECs when you mention it the first time (see page 2, line 60).

o   In general, the images are very small and the stained structures a difficult to detect. A scale bar in the figures (1a – e; 4b, e, and f) would be helpful.

o   Fig. 4b. The gastrocnemius muscle is a skeletal muscle and consists of multinucleated muscle fibers but not of individual muscle cells. Laminin stains the basement membrane of the muscle fibers, as can be seen in Fig. 4b. Please clarify it in the figure legends. What was the section plane of the muscle?

Reviewer 2 Report

Comments and Suggestions for Authors

1.    Authors should offer a contextual explanation of the canonical significance of Trim2's role.

2.    In Figure 1A, the variability in the percentage of CD68 is notable. Is there a specific reason behind this variability?

3.    In Figure 2D and EF, is there a discernible difference between Cuffed wild-type (wt) and Cuffed Trim2-/- samples?

4.    In Figure 2D, there appears to be no distinguishable difference in CD68 levels between non-cuffed wild-type (WT) and Trim-/- samples. However, in Figure 2A, a difference is evident. Shouldn't the non-cuffed condition represent the normal baseline?

5.    As the authors discussed, in Figure 3E, there is a increase observed in PVEGFR2 at the 1175 position, which may coincide with an decrease in phosphorylation at other sites, subsequently leading to decrease in eNOS activity. It is advisable for the authors to elucidate which specific tyrosine phosphorylation sites are decreasing, as this might correlate with the decrease in eNOS activity.

6.    As the authors demonstrated no change in blood flow in TRIM 2-/- mice following hind limb ischemia, can they provide insights into the potential reasons behind the observed increase in blood capillaries as shown in Fig 4C.

Comments on the Quality of English Language

English language can be improved 

Round 2

Reviewer 2 Report

Comments and Suggestions for Authors

Your responses to the queries raised during the review process were thorough and satisfactory. It is evident that you have diligently addressed all concerns and made necessary revisions to improve the quality and clarity of the manuscript